# System Operation of Regional UTM in Taiwan

**Chin E. Lin [1], Pei-Chi Shao [2,\*] and Yu-Yuan Lin [3]**

[1]  UAV Center, Chang Jung Christian University, Tainan 711, Taiwan; chinelin@mail.cjcu.edu.tw
[2]  Department of Aviation & Maritime Transportation Management, Chang Jung Christian University, Tainan 711, Taiwan
[3]  Telecommunication Lab., Chunghwa Telecom Co, Ltd., Taoyuang 300, Taiwan; stevenlin@cht.com.tw
**\***  Correspondence: pcshao@mail.cjcu.edu.tw

**Abstract:** The hierarchical unmanned aerial systems (UAS) traffic management (UTM) is proposed for UAS operation in Taiwan. The proposed UTM is constructed using the similar concept of ATM from the transport category aviation system. Based on the airspace being divided by 400 feet of altitude, the RUTM (regional UTM) is managed by the local government and the NUTM (national UTM) by the Civil Aeronautical Administration (CAA). Under construction of the UTM system infrastructure, this trial tests examine the effectiveness of UAV surveillance under 400 feet using automatic dependent surveillance-broadcast (ADS-B)-like on-board units (OBU). The ground transceiver station (GTS) is designed with the adoptable systems. In these implementation tests, five long-range wide area network (LoRa) gateways and one automatic packet reporting system (APRS) I-Gate are deployed to cover the Tainan Metropolitan area. The data rates are set in different systems from 8 to 12 s to prevent from data conflict or congestion. The signal coverage, time delay, data distribution, and data variance in communication are recorded and analyzed for RUTM operation. Data streaming and Internet manipulation are verified with cloud system stability and availability. Simple operational procedures are defined with priority for detect and avoid (DAA) for unmanned aerial vehicles (UAVs). Mobile communication and Zello broadcasts are introduced and applied to establish controller-to-pilot communication (CPC) for DAA. The UAV flight tests are generally beyond visual line-of-sight (BVLOS) near suburban areas with flight distances to 8 km. On the GTS deployment, six test locations examine communication coverage and effectiveness using ADS-B like OBUs. In system verification, the proposed ADS-B like OBU works well in the UTM infrastructure. The system feasibility is proven with support of receiving data analysis and transceiver efficiency. The trial test supports RUTM in Taiwan for UAV operations.

**Keywords:** hierarchical UAS traffic management (UTM); Automatic dependent surveillance-broadcast (ADS-B) like; relay gateways; detect and avoid; transceiver efficiency; data variance

## 1. Introduction

The Civil Aeronautical Administration (CAA) Taiwan legislated unmanned aircraft system (UAS) regulations on April 25, 2018 [1], and will open to the public for legal use on March 31, 2020. The regulations constrain the airspace of 400 feet above ground level (AGL) being regularly managed for small unmanned aerial vehicles (UAVs) flying at low speed. UAVs flying below 400 feet are authorized to local government for management; while those above 400 feet are controlled and monitored by the CAA. UAS traffic management (UTM) was studied by the National Aeronautics and Space Administration (NASA) to construct a feasible and effective system to assist UAS operation for aviation safety in 2015 [2]. The UTM system concept aims to incorporate small UAVs at low altitude to large mission UAVs in the integrated airspace. Under similar requirements, the hierarchical UTM is proposed and constructed in Taiwan [3] using the similar concept of air traffic management (ATM)

based on communication, navigation and surveillance (CNS) for UAVs. It is established by 400-feet segmentation of regional UTM (RUTM) and national UTM (NUTM) for management [3].

The National Airspace System (NAS) was developed for air traffic control (ATC) infrastructure in the 1960s. ATC insures air traffic clearance in the NAS [4] to maintain flight separation and safety for general category air transportations. ATM implements ATC into global civil aviation management. Based on the NAS, CNS are the main functions to link the aviation system in conjunction with ATC ground facilities and procedures. Flight surveillance is the key function of ATM [5]. Flight surveillance is also focused in the proposed UTM.

Initially, flight surveillance relied on the secondary surveillance radar (SSR) to track aircraft and reply to the ATM center by Mode S communication. The Communication, Navigation and Surveillance with Air Traffic Management (CNS/ATM) program introduced satellite technologies into ATM in 2010. The Global Positioning System (GPS) and Satellite Communication (SATCOM) improved greatly in CNS for ATM. Under this environment, the automatic dependent surveillance-broadcast (ADS-B) enforces aircraft position in broadcasting to vicinity aircraft and ground ATC centers with seamless communication [6].

In the developing UTM, an ADS-B like communication infrastructure is proposed by introducing 4G/LTE (long-term evolution), LoRa (long-range wide area network) and APRS (automatic packet reporting system) for small UAV (sUAU) [7]. The ADS-B like on-board unit (OBU) downlinks the UAV surveillance data to the UTM cloud. ADS-B like system in UTM is similar performance to ADS-B in ATM under the CNS concept.

In air surveillance, UAVs are not detectable by radar like manned aircraft. ADS-B shall be a feasible method to develop. To UAS, 4G/LTE or 5G are two systems with availability. Mobile communication is the most affordable communication system to adopt, for its wide area deployment. G. Orrell et al., presented an ADS-B like system concept using 4G/LTE for sUAS surveillance in low altitude [8]. The 4G/LTE is adaptable onto UAVs either cell phones or modules. However, 4G/LTE will not guarantee for those flying near or above 400 feet [9,10].

A command and control (C2) mechanism for large UAVs in high-altitude and long-range operations was proposed [11]. It solves CNS problems for high altitude and long endurance UAS operations. C. Ramsey introduced a small ADS-B device by "uavionix" for UAVs. It uses commercial frequency of 1090 MHz and 978 MHz for UAVs. However, radio stations for ADS-B are not designed to provide coverage down to low altitude [12]. The UAS providers are not affordable to construct a new infrastructure for UAS. J. Scardina presented the use of ADS-B for general categories of commercial air transportation systems [6] by FAA and ICAO. Under very limited ADS-B code sources, it is difficult to broadly apply ADS-B to all types of UAVs, especially those small ones flying below 400 feet. In addition, since a lot of UAV ADS-B (978 MHz) is used in general aviation under 500 feet, the growth of ADS-B used in sUAS may result in significant over density in low-altitude airspace [13]. In the developing hierarchical UTM system, the ADS-B like communication is designed to adopt devices [7] with high reliability, light weight, low cost and wide coverage through ground transceiver station (GTS) deployment. Other than 4G/LTE, the LoRa and APRS require constructing an exclusive communication GTS to relay radio surveillance from UAVs into the UTM Cloud [3,7]. A similar application of cellular network is proposed by R. Yeniçeri, et al. [10] to construct a centralized UTM. From the reference works, the proposed ADS-B like GTSs relay all UAV surveillance data into the UTM cloud, and distribute them to regional UTM (RUTM) for local governments, and to national UTM (NUTM) for the CAA.

In UTM, similar to ATM, it is necessary to establish an effective radio link for controller-to-pilot communication (CPC) for any emergency. Since mobile communication 4G is most accepted for person to person (P2P), the proposed UTM selects mobile communication as a standard communication means. In addition, Riley introduced a new "Zello" created by Gavrilov to offer additional feature of a two-way radio for general use [14,15]. It is designed to chat like a walkie-talkie on special handsets or using mobile communications. It is convenient to set up individual pilot groups for flight operations. The UTM controller can broadcast voice command to pilot groups as one option of CPC.

By setting up the GTS for LoRa and APRS, trials have been made continuously to examine the effectiveness of the proposed surveillance communication with reporting data. The UAV on-board units (OBU) broadcast data (Tx) to GTS, and receive (Rx) into the UTM cloud. It will need to analyze the transceiver efficiency and proceed data statistics. In the trial tests, since the data quantity is not plenty at present, the data statistics would need to examine its feasibility and distribution variance. This property may fit with method based on the binominal distribution [16]. The signal gains of APRS and LoRa are compiled with the binomial experiment for 4 properties as: (1) The experiments consist of a sequence of *n* identical trials. (2) Two outcomes are possible on each trial, where one outcome is accepted as a *success* and the other outcome as a *failure.* (3) The probability of a success, denoted by *p*, does not change from trial to trial. Consequently, the probability of a failure, denoted by 1–*p*, does not change from trial to trial. (4) The trials are independent. In the binomial experiments, the number of successes occurring in the *n* trials for the APRS and LoRa signal depend upon the nature of discrete variables. Rx are measured by 5 ground transceiver stations and Rx means the successful number in probability as denoted by p. The variance of binomial experiments means the stability of the signal and consistency.

According to the Taiwanese UAS Regulation, the proposed UTM trial run is tested in Tainan RUTM under the approval of the Tainan City Government. The test tries to verify the construction of UTM infrastructure with the performance of ADS-B like communication for Tainan RUTM. From this standpoint, further tests and verifications can push forward to elevate a reliable operation of the UTM infrastructure.

## 2. Unmanned Aerial Systems Traffic Management (UTM) System Concept

In the European Union (EU), U-Space is proposed as UTM from NASA to construct an airspace for UAS traffic in low altitude. UTM declares 400 feet as a minimum level to allow small UAS (sUAS) to operate [2]; while in the U-Space, the uncontrolled Class G airspace under 700 feet is suggested for UAS operation [17,18]. Labib [17] proposed multi-layer concept for UAS traffic in the urban areas from 0 to 700 feet. UAV path planning on the designated traffic layer becomes an important task in analysis before take-off. Barrado proposed the similar concept in very low-level (VLL) airspace to create corridors for UAS. In the VLL airspace, X, Y, Z volumes are designed for various operations. Since the VLL is usually empty to manned aviation except some emergency flights, based on the U-Space traffic management system design, multi-layer airways will be deployed for dense UAS traffic. Under UTM airspace planning [2], 400 feet is legally released to the public in Taiwan [1]. Constraints are pronounced for no-fly zones (NFZs), high population areas, near freeways or railroads. Considering the downwash of multi-rotors, UAV vertical separation is not applied in UTM. The proposed hierarchical UTM will construct a flight environment for UAVs, especially below 400 feet for regional UTMs.

To enforce clear traffic management for all UAVs, surveillance is the most important technology to adopt. Referring to ADS-B in the manned aircraft, the hierarchical UTM focuses on the development of ADS-B like technology to offer UAVs with complete transparency to UTM control during flights.

Following the ATM surveillance, as shown in Figure 1, a similar surveillance concept for UTM is proposed using ADS-B like mechanism. The surveillance data from ATM is fed into ATC center via the aeronautical telecommunication network (ATN) [19]; while those from UAVs are connected into the UTM cloud through the Internet.

In CNS/ATM, the ADS-B ground stations are deployed for a seamless coverage from station to stations for transportation category aircraft [19]. SITA and ARINC invested most ADS-B ground stations globally for air transportation. But those ADS-B ground stations may not offer services to UAVs [8,12] now or in the future. The proposed UTM using ADS-B like communication constructs specific GTS to relay UAV surveillance data from low altitude flights. The LoRa gateway and APRS i-Gate are selected as GTS in the UTM implementation. A UTM cloud is established to collect UAV surveillance data from ADS-B like infrastructure. Figure 2 shows the ADS-B like GTS deployment plan for UTM in Southern Taiwan. The red (smaller) circles of 15 km radius are LoRa gateways;

while the blue (larger) circles of 40 km radius are APRS i-Gates. In Figure 2, the rectangle area is Tainan City, which is running an integrated pilot program (IPP) for RUTM trials under this project. In the preliminary tests of GTSs, five LoRa gateway stations and one APRS i-Gate stations are planned to construct and offer sufficient coverage to Tainan City for RUTM. APRS is quite popular among HAMs radio amateurs, there are more than 200 APRS gateway stations in Taiwan using 144.64 MHz. Although 4G/LTE may offer good services with much wider range but less altitude coverage above 400 feet [10,13]. Figure 3 shows the altitude coverage for LoRa Gateways. It is expected of theoretical coverage above ground to 400 feet and then 20,000 feet. A similar concept is designed in APRS with higher coverage. For civil UTM development, fixed wing UAVs are expected to fly 4000 to 12,000 feet; while the rotor wing UAVs are flying below or near 400 feet. The selected LoRa and APRS are distinct from mobile and other communication, the UTM project needs to distinguish GTS for the UTM operation only. In the future, a legal APRS frequency for UTM, other than 144.64 MHz, will be applied and authorized by National Communication Commission (NCC).

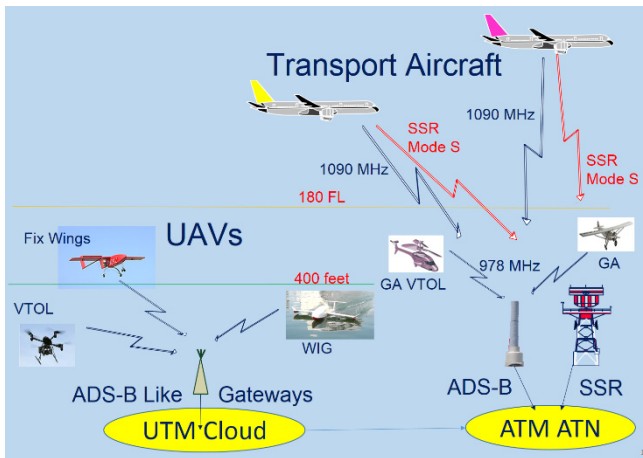

**Figure 1.** Surveillance concept for unmanned aerial systems traffic management (UTM) to air traffic management (ATM).

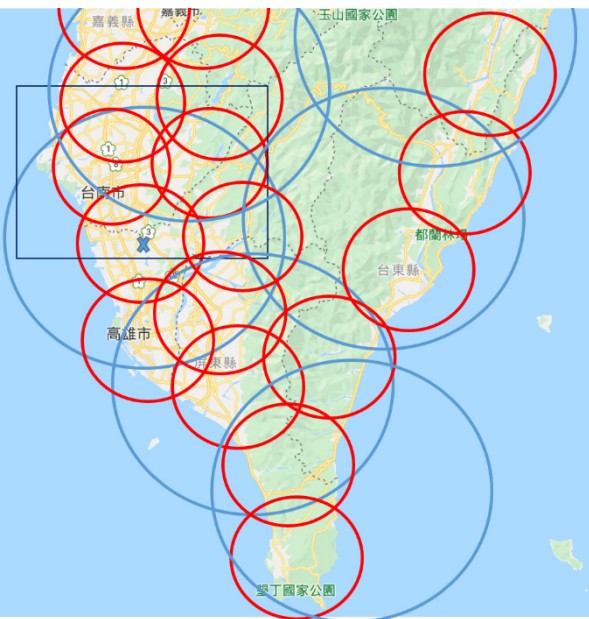

**Figure 2.** Automatic dependent surveillance-broadcast (ADS-B) like ground transceiver station (GTS) deployment in Southern Taiwan (red small circle for long-range wide area network (LoRa), blue large circle for automatic packet reporting system (APRS), CJCU at **X**, where rectangle enclosing Tainan City).

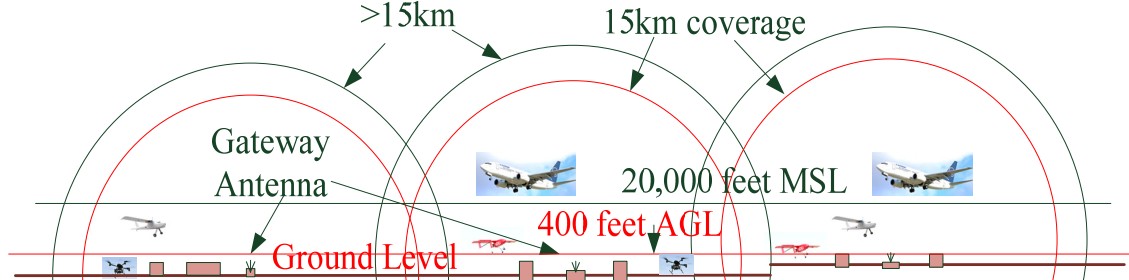

**Figure 3.** Theoretical altitude coverage of LoRa Gateway (similar to APRS with higher coverage).

From the system operation concept, the proposed hierarchical UTM is organized by RUTM and NUTM [3]. All UAV surveillance data will be relayed via GTS into the UTM cloud. The ADS-B like infrastructure is constructed and deployed in Tainan City for RUTM operation. All UAVs will be classified into small rotor wings to fly below 400 feet, or larger fixed wings to fly above 400 feet. All UAVs will fly under surveillance in RUTM to NUTM. An additional concept in Figure 4 expresses the possible connection from UTM to ATM via the air navigation service provider (ANSP) to re-broadcast UAV surveillance data to NAS aircraft. ADS-R frequency 1090 MHz to air transportation aircraft or 978 MHz to general aviation aircraft are re-broadcasted. The traffic alert and collision avoidance (TCAS) together with detect and avoid (DAA) by active avoidance shall be maneuvered by aircraft pilots [20,21].

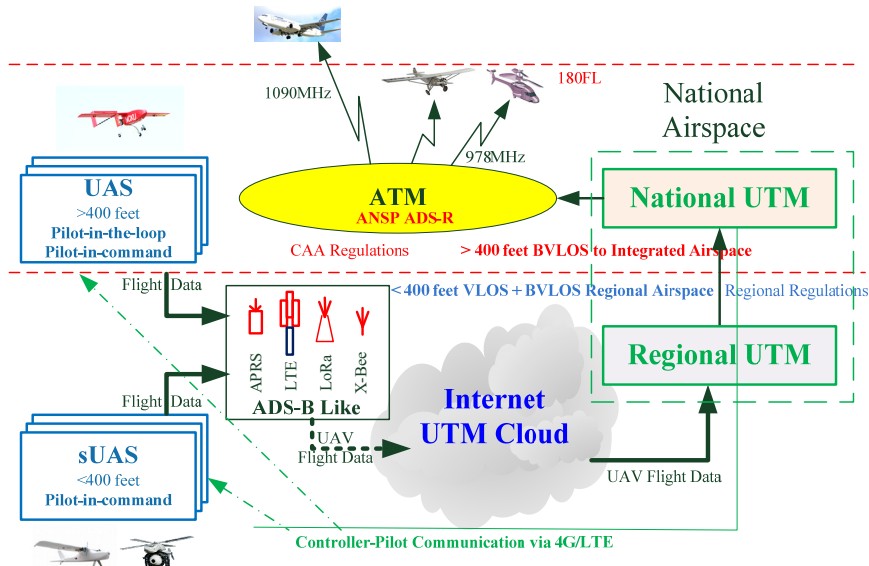

**Figure 4.** The hierarchical UTM in Taiwan.

In the proposed hierarchical UTM, the surveillance data is designed with 90 Byte data packet to contain: [Heading (5); UAV (6); Pilot (6); Lat. (9); Long. (10); Alt. (4); 6 degrees of freedom (DoF) (p, q, r, $\alpha$, $\beta$, $\gamma$) (36); V (6); A (6); Tail (2)], where 6 DoF (degrees of freedom) flight data are included. A special heading "@@@**" and tailing "##" are selected to distinguish from other users, especially the HAMs in APRS. The reason for the 90 byte data packet concerns the flight information acquisition for further analysis, such like flight operation quality assurance (FOQA) in the civil aviation system. Rodriguez, et al. [22] and Mattei, et al. [23] paid attention to the flight control awareness and tried to report 6 DoF via ADS-B communication. With big data collection, UTM FOQA would definitely be valuable to offer great contribution to UAV/UAS safety assessment in the future.

## 3. UTM System Construction

### 3.1. System Infrastructure

Figure 5 shows the proposed hierarchical UTM system with information flow infrastructure. A UTM cloud is established to receive all surveillance data from UAVs using the proposed ADS-B Like communication [7]. The UTM will be operated under a similar concept of ATM for CNS. In the UTM center, controllers will intervene in pilot operations for flight surveillance. Services from UTM include pilot log-in/out, UAV log-in/out, flight planning, CPC, DAA [24], full surveillance, ground control data backup, incident record, accident lock-on and rescue, etc. The prototype UTM center is constructed in CJCU. LoRa and APRS ground transceiver stations are deployed in Tainan City as shown in Figure 2.

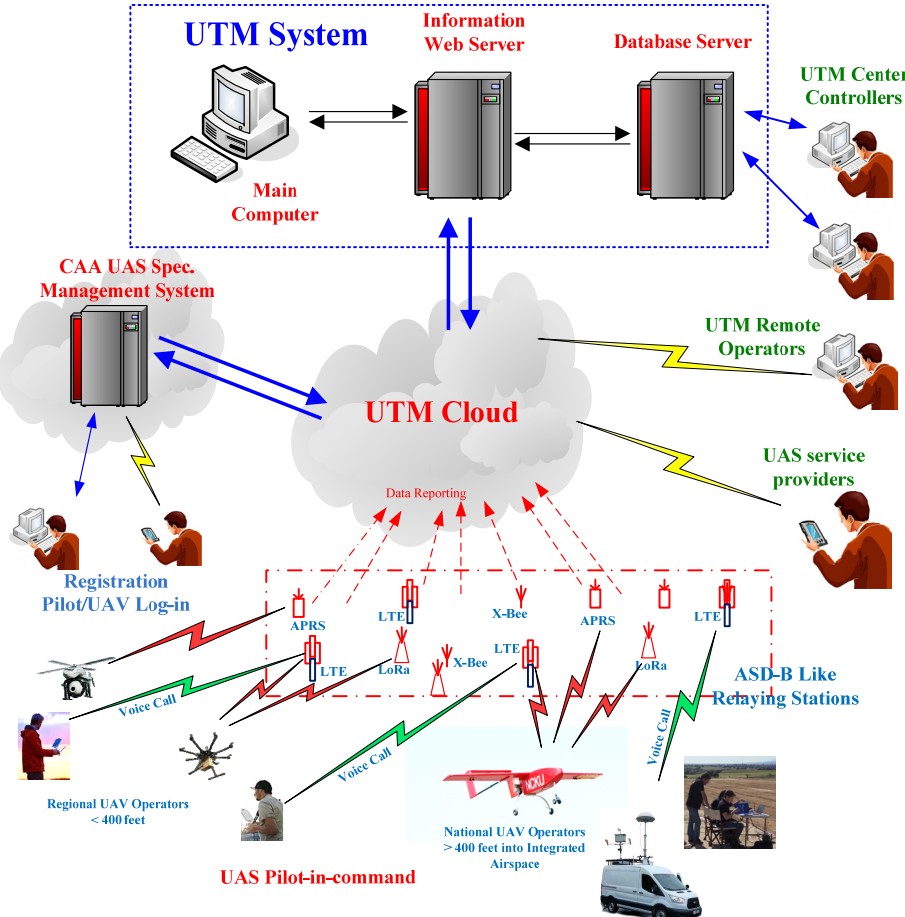

**Figure 5.** The proposed hierarchical UTM system.

Based on the proposed concept, the UTM system structure is designed and constructed with application program interface (API) server and web server as shown in Figure 6. In the developing phase, the UTM combines Tainan RUTM and NUTM for trials and demonstrations. Similar to ATM, a controller-to-pilot communication (CPC) in RUTM using mobile communication and Zello broadcast is established.

The CAA develops a "UAS Management Information System" (MIS) to establish a database of qualified pilots, registered UAV, no-fly zone (NFZ) and all related information in a complete system. The proposed UTM shall connect to MIS for pilot, UAV and flight route reconfirmation. A process of API to MIS is connected before flight plan approvals, as shown in Figure 7. Pilot cell phone shall be registered together with pilot ID and UAV ID. Pilots shall select or create a Zello group for further communication. The CAA is working on legal process to set up databases of (1) pilots, (2) UAVs

and (3) NFZs. For all pilots, written test and flight test are required to pass the minimum level for a pilot license. All UAVs less than 25 kg need to register, and those heavier than 25 kg need type certification [1]. All no-fly zones (NFZ) including restrict areas and forbidden areas will be pronounced by CAA and local governments. All UAV flight plans shall avoid and by-pass the NFZ.

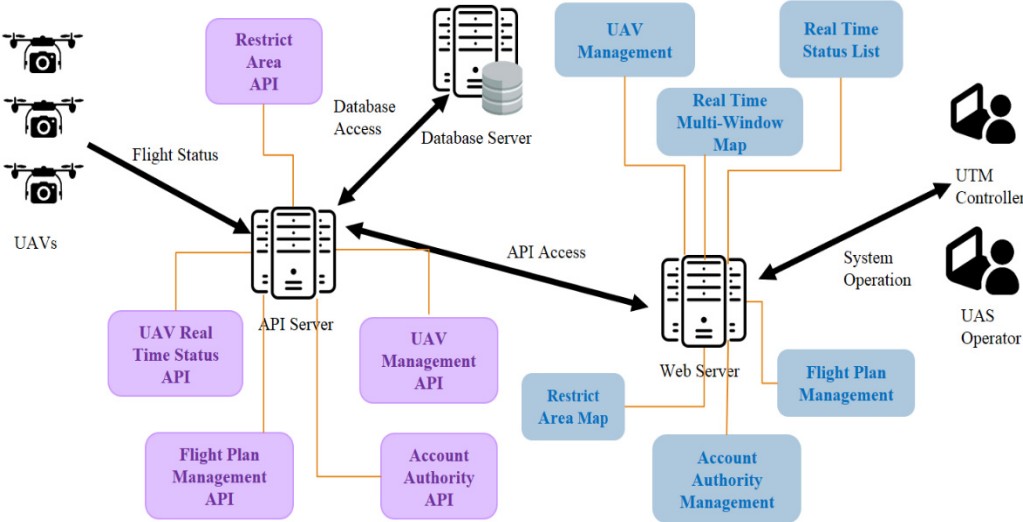

**Figure 6.** The proposed UTM system structure for tests.

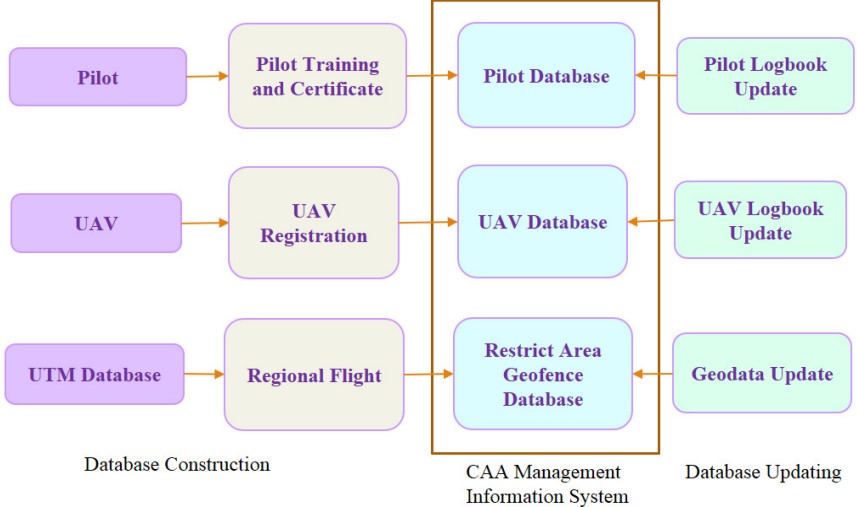

**Figure 7.** UTM application program interface (API) connection to Civil Aeronautical Administration (CAA) Management Information System (MIS).

*3.2. ADS-B Like On-Board Unit (OBU)*

The UTM development has moved forward into flight tests after the ADS-B Like on-board units (OBU) being ready for use from a preliminary design and fabrication [7,22]. Two types of ADS-B Like OBUs have completed flight tests to report 90 byte surveillance data onto the UTM cloud. Figure 8 shows the LoRa OBU and APRS OBU. In this phase of development, a 4G cell phone is directly selected for test. CHT will offer 5G module in future tests.

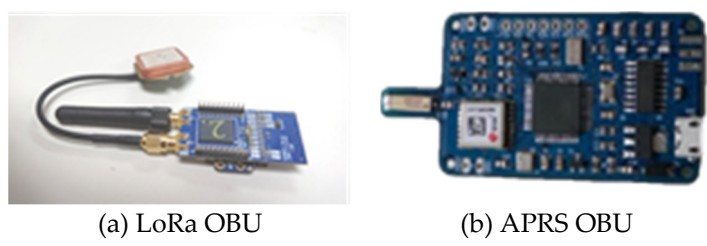

(a) LoRa OBU           (b) APRS OBU

**Figure 8.** ADS-B like on-board unit (OBU), LoRa and APRS.

The flight test fleet includes 550, DJI P4P, 930 types and 380 quad-rotors with an additional one M600 hexa-rotor. Some of them are shown in Figure 9 with OBU mountings. APRS, LoRa OBU and 4G cell phone are ready for tests. For DJI P4P, it needs special mounting fixture design; while others may directly affix to UAVs according to their OBU axial marks. Since the UTM commands UAVs to report 90 Byte data including 6 DoF, heading direction is the key setting to recognize the UAV flight awareness.

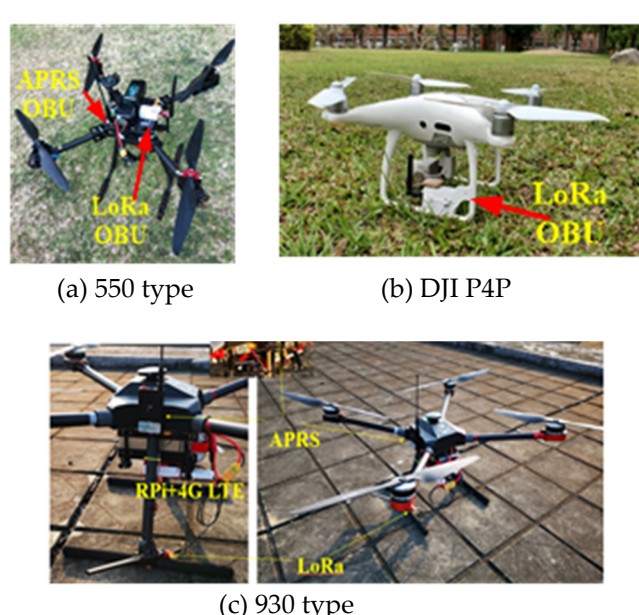

(a) 550 type           (b) DJI P4P

(c) 930 type

**Figure 9.** OBUs mounting on UAVs for flight tests.

## 4. System Verification

In the development of the hierarchical UTM, the adopted technology should be established and verified by flight tests. Tainan Regional UTM (RUTM) is designed and constructed based on the proposed ADS-B like surveillance infrastructure. The GTS should be deployed to install the LoRa Gateways and APRS i-Gates for trials.

After the GTS installations, there are immediate flight tests near the GTS. Additional trials are also carried in some remote areas in Tainan City. All flight data shall be collected in the RUTM center. The data statistics in the trial flights is analyzed to examine the ADS-B like OBU in its transceiver capability and data reliability.

### 4.1. Long-Range Wide Area Network (LoRa) Ground Transceiver Stations (GTS) Deployment and Tests

On the construction of hierarchical UTM, a coverage survey on ground transceiver stations (GTS) in Tainan City using APRS and LoRa is planned and executed to affix the gateways or i-gates on available CHT buildings. In Tainan, five LoRa GTS (in small red circles) will be roughly enough to cover whole city territory, in addition, a APRS GTS (in large blue circle) is also installed on the CJCU campus for test. It is shown in Figure 10 which may refer to Figure 3 of the conceptual ADS-B Like infrastructure design.

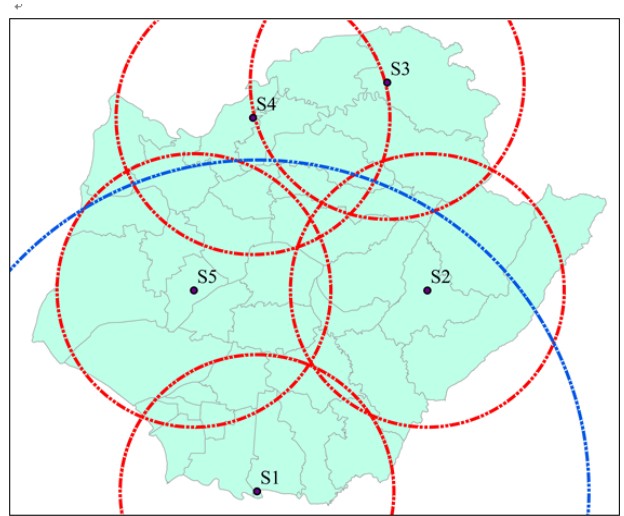

**Figure 10.** LoRa and APRS GTS coverage in Tainan City.

Five LoRa GTSs are installed to deploy the ADS-B Like coverage for Regional UTM (RUTM). The S1 station at bottom of Figure 10 refers to CJCU (S1) station, while the other four GTSs are located at Yu-Jing (S2), Bai-Her (S3), Yen-Shui (S4) and Xi-Gang (S5). In preparation, legal procedures have been completed: (1) six qualified pilots with CAA license to perform flights, (2) five UAVs being registered with liability insurance for tests, (3) airspace and flight plan application and approval by Tainan City Government to fly under 400 feet, (4) all UAVs under ADS-B like surveillance by RUTM.

After GTS installation, flight tests are immediately made near the GTS for its performance with OBU. The received signal strength indicator (RSSI) is measured from GTSs. The RSSI is observed from −90 to −120 dBm. The test flies at about 6 m/sec velocity. The OBU is set at report period of 8 s, according to CAA rule. The transceiver efficiency ($\eta$) is calculated following the transmitted data (Tx) from OBU and received data (Rx) in UTM cloud. "Time + $\Delta$t" indicates the flight time (hour + minute) and $\Delta$t indicates flight duration in test.

The first GTS station is located on the CJCU campus with LoRa GTS and APRS GTS for initial tests to report surveillance data to UTM Center in Figure 11. Test data of LoRa GTS is shown in Table 1. Its flight data is recorded in UTM cloud. This flight uses a Hexa rotor M600.

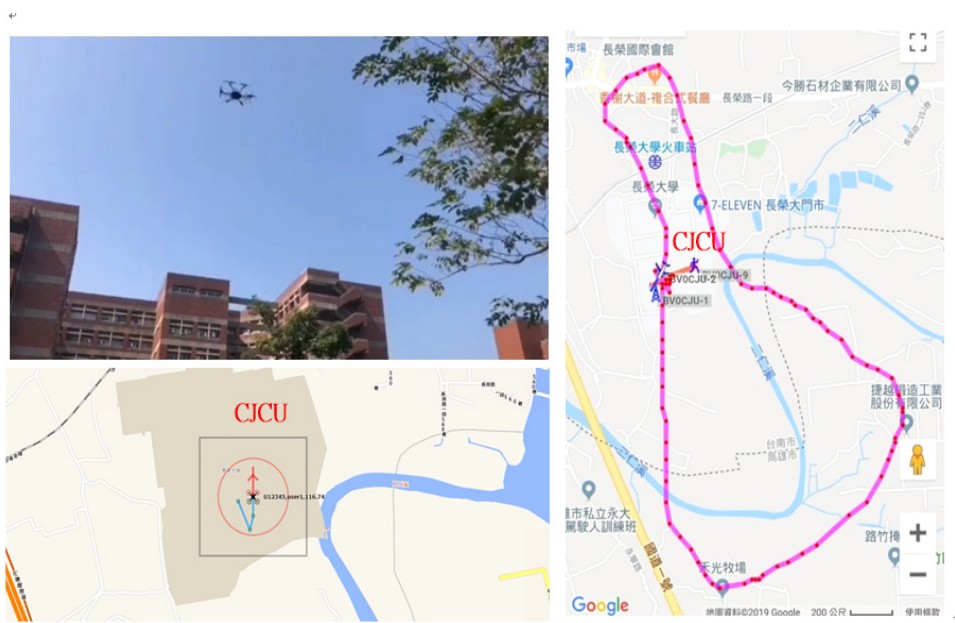

**Figure 11.** APRS flight test on CJCU campus with real-time tracking on UTM display and its flight record.

**Table 1.** Test data at CJCU (S1).

| GTS S1 | LoRa | UAV | M600 |
|---|---|---|---|
| Antenna | 71 m MSL | Fly Alt | 60 m AGL |
| RSSI | −95 dBm | Distance | 1~7 km |
| Time + Δt | 1515+21 | | |
| Tx | 159 | η = Rx/Tx | 0.94 |
| Rx | 156 | BD | 0.17 |
| Loss | 3 | **Variance** | **5.63** |

In Table 1, the effective transmission data is analyzed by binomial distribution (BD) as: $\binom{n}{x} p^x (1-p)^{x-n}$, where *n* and *x* are data number sent or received, $n = Tx$, $x = Rx$, *p* is the percentage of success [16]. The data variance of LoRa GTS can be calculated by Variance = $np(1-p)$. The "**Variance**" refers to the stability and reliability of GTS using the ADS-B like OBU.

After LoRa GTS installation, the Yu-Jing station (S2) is shown in Figure 12 with immediate test data in Table 2. Similar work has been done for the other LoRa GTSs being installed and tested to get analytical data from Tables 3–5. These flight tests use a Quad-rotor 550 for easy carry to operate outside campus. This deployment plan will try to cover Tainan City for the first verification trial. The LoRa OBU period is set at 8 s.

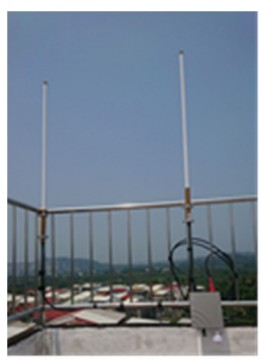 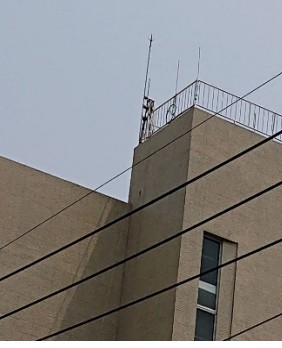 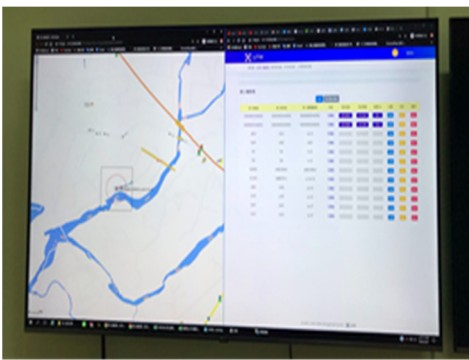

(a) Roof top     (b) Antenna setting     (c) Flight test on UTM display

**Figure 12.** LoRa GTS site S2 and verification test on UTM display.

**Table 2.** Test data at Yu-Jin (S2).

| GTS S2 | LoRa | UAV | DJI P4P |
|---|---|---|---|
| Antenna | 120 m MSL | Fly Alt | 50 m AGL |
| RSSI | −90 dBm | Distance | 5~8 km |
| Time + Δt | 1100 + 15 | | |
| Tx | 109 | η = Rx/Tx | 0.89 |
| Rx | 97 | BD | 0.12 |
| Loss | 12 | **Variance** | **10.68** |

**Table 3.** Test data at Bai-He (S3).

| GTS S3 | LoRa | UAV | 550 |
|---|---|---|---|
| Antenna | 800 m MSL | Fly Alt | 50 m AGL |
| RSSI | −100 dBm | Distance | 6~7 km |
| Time + Δt | 1105 + 06 | | |
| Tx | 34 | η = Rx/Tx | 0.41 |
| Rx | 14 | BD | 0.14 |
| Loss | 20 | **Variance** | **8.24** |

**Table 4.** Test data at Yen-Sui (S4).

| GTS S4 | LoRa | UAV | 550 |
|---|---|---|---|
| Antenna | 40 m MSL | Fly Alt | 50 m AGL |
| RSSI | −100 dBm | Distance | 6~7 km |
| Time + Δt | 1055 + 09 | | |
| Tx | 64 | η = Rx/Tx | 0.88 |
| Rx | 56 | BD | 0.15 |
| Loss | 8 | **Variance** | **7.00** |

**Table 5.** Test data at Xi-Gan (S5).

| GTS S5 | LoRa | UAV | 550 |
|---|---|---|---|
| Antenna | 42 m MSL | Fly Alt | 55 m AGL |
| RSSI | −90 dBm | Distance | 5~8 km |
| Time + Δt | 1515 + 06 | | |
| Tx | 98 | η = Rx/Tx | 0.94 |
| Rx | 92 | BD | 0.17 |
| Loss | 6 | **Variance** | **5.63** |

The GTS S3 has some interference from 4G BTS about 20 m range. From Tables 1–5, these five LoRa GTSs are deployed for RUTM operation in Tainan City. Figure 13 shows the flight surveillance at UTM Center after GTS being installed. The trial data in binominal distribution (BD) are varying. However, following the transceiver efficiency, the GTS would be examined with their performance efficiency and availability according the resulting "**Variance**". From Tables 1–5, S5 = 5.63, S4 = 7.00, S3 = 8.24, S2 = 10.68, S1 = 5.63. S2 seems to be poor in data transceiver performance. It needs some additional trial tests to check the antenna setting and environment affection. S1 and S5 are good GTS sites to work with the UTM operation. By now, it is difficult to assert what range will be suitable for GTS by analyzing their data "**Variance**".

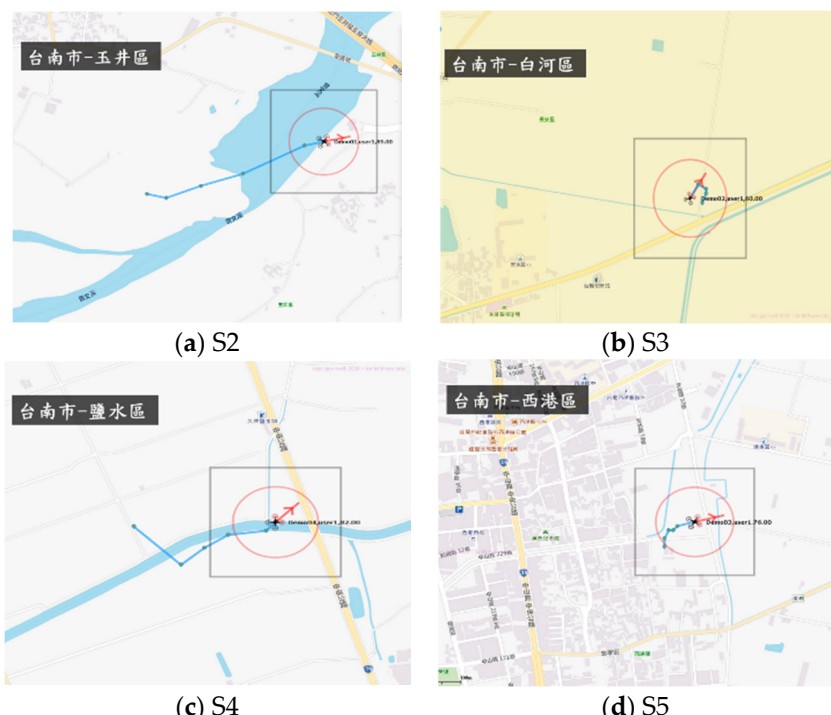

(**a**) S2  (**b**) S3

(**c**) S4  (**d**) S5

**Figure 13.** Immediate test after GTS site installation.

### 4.2. LoRa Verification Flight Tests

After finishing GTS deployment for Tainan RUTM, a full verification flight test is carried out at different locations in the northern part of Tainan City. Five GTSs are checked with data transceiver operation. The verification flight tests try to examine the performance of RUTM using LoRa and APRS by the following conditions: (1) UAV flight altitude, 30 m~90 m AGL, (2) Receiving surveillance data from which GTS, (3) Estimate UAV distance to GTSs, (4) RSSI −90 to −120 dBm, (5) Signal data transceiver efficiency and data variance, (6) Observing environment interference. The UAVs used for tests are a Quad-rotor 550 and a DJI P4P. The OBUs are mounted on board. The LoRa OBU is exclusively designed for independent use with battery power, inertia measurement unit (IMU) and compass without any connection from the UAV. A special mounting fixture is fabricated to fix on DJI P4P. For other types of UAV, such as 550, either OBU by APRS or LoRa can be directly mounted. APRS OBU is designed with no IMU and compass. The APRS OBU connects to Pixhawk to obtain 6 DoF flight data [25]. This OBU is not applicable to those enclosed designs, such as DJI products.

Figure 14 shows the test locations and GTS in Tainan City. The test sites (T1 to T6) are selected a little farther away from GTS or near the coverage boundary. The distances from test sites to GTSs are marginal to farther than 15 km. Its purpose is to find the coverage redundancy for GTSs. The flying altitudes are recorded from 30 m above the ground and as high as to 90 m (AGL). The CJCU (S1) GTS is located farthest during these tests in northern Tainan City.

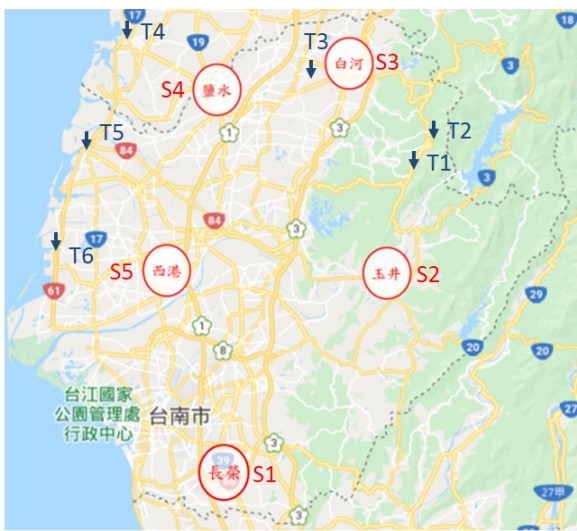

**Figure 14.** GTS deployment and site test, S1~S5 and T1~T6 in the northern part of Tainan City.

In Table 6, the flight test sites are located a little far from 5 LoRa GTSs to test its communication capability and coverage larger than 15 km. Referring from Tables 1–5, since the receiving data are poor compared from Tables 1–5, the binominal distribution and data variation appear less meaningful to data statistics. However, the data "**Variance**" in Tables 1–5 are valuable as an important index for reference. Table 6 presents ADS-B like OBU capability and coverage of LoRa GTSs.

In Table 6, the test sites are selected critical to ADS-B Like performance. The records of shaded parts are complete to cover all T1~T6 test data. By analysis, it is:

(1)　T6 can be received by S1 and S5 in good condition;

(2)　T5 can be received by S5 and S4 in poor condition;

(3)　T4 can be received by S4 but in poor condition;

(4)　T3 can be received by S3 in good condition;

(5)　T2 can be received by S2 in good condition but with interference;

(6)　T1 can be received by S5 in good condition.

**Table 6.** Flight test data analysis.

| Test Site | T1 | T2 | T3 | T4 | T5 | T6 |
|---|---|---|---|---|---|---|
| Time + Δt | 1053 + 09 | 1131 + 13 | 1319 + 08 | 1437 + 02 | 1516 + 04 | 1552 + 6 |
| Tx | 65 | 95 | 58 | 12 | 28 | 42 |
| Elevation m | 300 | 600 | 60 | 10 | 11 | 5 |
| Fly Alt m | 95~120 | 35~60 | 30~60 | 30~70 | 40~70 | 30~90 |
| **GTS CJCU (S1) Antenna top at 71 m MSL, Data Variance = 5.63** | | | | | | |
| **RSSI** | X | X | X | X | X | **−110** |
| Distance km | 43.5 | 46.5 | 48.5 | 60.1 | 43.8 | 20.7 |
| η = Rx/Tx | Too far | Too far | Too far | Too far | Too far | 0.67 |
| **GTS Yu-Jin (S2) Antenna top at 120 m MSL, Data Variance = 7.00** | | | | | | |
| **RSSI** | X | X | X | X | X | X |
| Distance km | 14.1 | 16.4 | 23.3 | 46.2 | 38.2 | 38.5 |
| η = Rx/Tx | X | X | Too far | Too far | Too far | Too far |
| **GTS Bai-He (S3) Antenna top at 80m MSL, Data Variance = 8.24** | | | | | | |
| **RSSI** | **−106** | **−100** | **−100** | **−113** | X | X |
| Distance km | 13.3 | 12.8 | 3.2 | 27.6 | 30.6 | 39.8 |
| η = Rx/Tx | 0.02 ** | 0.95 ** | 0.57 | 0.33 | Too far | Too far |
| **GTS Yen-Sui (S4) Antenna top at 40m MSL, Data Variance = 10.68** | | | | | | |
| **RSSI** | X | **−108** | **−110** | **−108** | **−105** | X |
| Distance km | 23.1 | 24.3 | 13.7 | 17.6 | 15.2 | 25.5 |
| η = Rx/Tx | Too far | Too far | 0.03 | 0.17 | 0.14 | Too far |
| **GTS Xi-Gang (S5) Antenna top at 42 m MSL, Data Variance = 5.63** | | | | | | |
| **RSSI** | **−110** | **−120** | X | **−118** | **−120** | **−93** |
| Distance km | 31.9 | 34.4 | 30.4 | 35.0 | 18.7 | 12.7 |
| η = Rx/Tx | 0.86 | Too far | Too far | Too far | 0.68 | 0.90 |

Notes: 1. Data from test location being "**too far**" from the GTS will be ignored. 2. X means GTS does not receive flight data. ** experienced with significant interference by 4G BTS. 3. GTS antenna top = ground elevation (MSL) + building height (AGL). 4. Test site height = site elevation (MSL) + flight altitude (AGL). 5. Time +Δt = Flight time + flight duration. Flight duration/8 s period = data count of each flight. 6. Received signal strength indicator (RSSI) = X means GTS cannot receive signals, test location too far to get signal too weak < −130 dBm. 7. Distance = Test site to GTS. Typical LoRa gateway covers 15 km radius. 8. Transceiver efficiency η = Data received (Rx)/Data transmitted (Tx) = Effective data numbers sent from OBU to UTM.

The flight tests verify the coverage, availability and reliability of the ADS-B like system infrastructure of LoRa GTS and OBU. During the flight tests, the APRS OBU is also affixed on UAV to send surveillance data. All surveillance data via APRS are clearly received by the UTM center. This part is separately presented [25].

During the tests, the receiving signal strength indicator (RSSI) of reporting data maintain to acceptable level of −90 to −120 dBm. Although the design specification of LoRa gateway guarantees a feasible limit of 15 km range, by tests, the coverage of LoRa GTS may be much farther than the 15 km specification. From other tests, LoRa GTS and APRS GTS have been verified with much longer coverage than their specifications.

In this preliminary test, since the locations are farther than the specification range, the transceiver efficiency is not acceptable at all. The radio interference from 4G BTS at Bai-He GTS (S3) affects radial 150° to 180°. Further flight tests are required to adjust the GTS antenna setting.

*4.3. Automatic Packet Reporting System (APRS) GTS Verification Test*

In UTM development, one APRS GTS is established on the CJCU campus and several APRS GTSs from HAMs are available. Referring to Figure 10, the large blue circle indicates the APRS coverage from CJCU GTS. A preliminary verification test is carried out to examine the received signal strength and transceiver efficiency (see Table 7). The test site is located 40 km from the CJCU campus at 184 m ground elevation. This flight tries to verify APRS transceiver efficiency, the flight duration is short.

**Table 7.** Test data at CJCU (S1) on APRS GTS.

| GTS S1 | APRS | UAV | 550 |
|---|---|---|---|
| RSSI | −100 dBm | | |
| Antenna | 71 m MSL | Period | 8 sec |
| | Test a | | |
| Time + Δt | 1455+05 | Distance | 37~42 km |
| Elevation | 184 m MSL | Fly Alt | 55 m AGL |
| Rx/Tx | 24/27 | η | 0.98889 |
| BD | 0.0031 | Variance | 0.2966 |
| | Test b | | |
| Time + Δt | 1515 + 04 | Distance | 33~43 km |
| Elevation | 184 m MSL | Fly Alt | 60 m AGL |
| Rx/Tx. | 17/17 | η | 1.0 |
| BD | 1.0 | Variance | 0.0 |

It was observed that APRS transceiver efficiency is very good at such a long distance. This verifies that APRS coverage is quite stable and reliable within its coverage. APRS is widely used by HAMs at 144.64 MHz. During the flight tests, the frequency is tuned to 144.61 MHz to avoid annoying interference from other users. However, some tests still have experienced with signal being covered up by nearby high power stations from 144.64 MHz users.

*4.4. System Redundancy*

In the UTM development, system redundancy is an important index for operation and performance. The UTM redundancy is twofold: (1) Single OBU, LoRa or APRS, overlapping coverage of multiple GTSs, (2) Multiple OBU, using LoRa or APRS, to receive higher data transceiver efficiency. In Table 6, five LoRa GTS deployments in Tainan City present quite feasible redundancy to each other to compose a complete surveillance system network for the UTM operation.

Additionally, the proposed multiple ADS-B Like infrastructure offers another redundant concept to UAS, as shown in Figure 15. Although 4G/LTE is expected to maintain higher connectivity levels [9], not all communication can be guaranteed with 100% reliability anywhere any time. Also, 4G/LTE or even future 5G, cannot guarantee the altitude coverage to 400 feet or beyond. The LoRa GTS and APRS GTS of the ADS-B Like infrastructure can offer much higher altitude coverage as shown in Figure 3. A communication redundancy is mandatory to enhance UAS reliability under surveillance using an ADS-B like communication infrastructure. In Figure 15, APRS is particularly recommended to large fixed wing UAVs to fly into NAS, as shown in Figure 4. From the trial test results, it is strongly recommended that fixed wing UAVs flying high altitude into remote mountain area would need to be equipped with LoRa and APRS OBUs to obtain higher reliability for UTM.

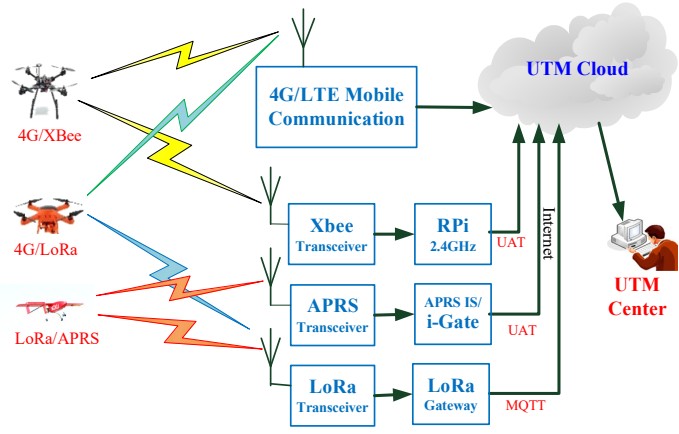

**Figure 15.** System redundancy using multiple ADS-B like system.

### 4.5. UTM Center

Under the proposed hierarchical UTM, a UTM center for Tainan RUTM is constructed for trial operation. This first UTM control center is located at CJCU. There are main servers, one HP Z8 and two PCs, to receive and process UAV data from the UTM cloud. Additional personal computers are installed to support UAV surveillance real time display on two projectors and two monitors. It is shown conceptually in Figure 16. Multiple computer design is capable of reducing the central processing unit (CPU) load and display output load in real time flight projection. Figure 17 shows the complete view of the UTM control center with two controllers in service.

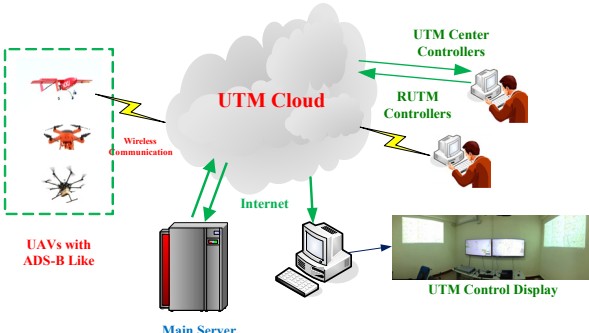

**Figure 16.** UTM Control Center.

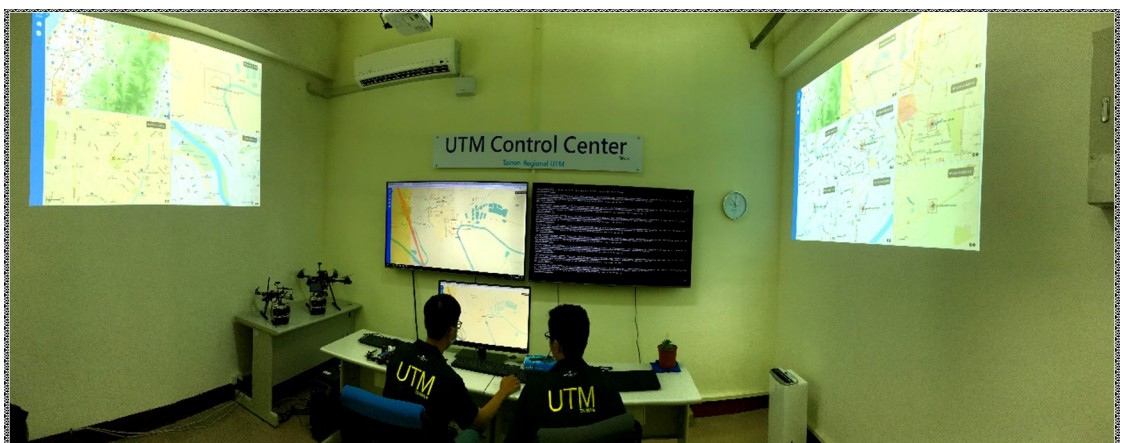

**Figure 17.** Regional UTM (RUTM) control center in full view.

### 4.6. Six Degrees of Freedom (6DoF) Surveillance and Data Suppression

The UTM surveillance data is designed with 90 Bytes from OBUs. The purpose of 90 byte data would require to contain not only position but also flight control with 6 DoF. The flight records will be valuable to support FOQA for UAVs, and will be as important as it in the air transportation system. The proposed UTM will collect UAV big data in the UTM cloud. 6DoF big data base will offer further operation analysis. The 90 Byte data string appears by data starting at @@@** and ending at ## to distinguish from other users.

2020-02-21 18:16:08 036 4.83@@@**U00005O0000523.23081N120.37173E0184-5.51 3.96-0.32-3.4 127.3-1.1 23.99 12.13## Sending to DroneLives!! [25].

Figure 18 shows the presentation of 6DoF during the UAV flight, where a bank angle to turn is shown.

In programming, the 90 byte data occupies transmitter load. The data needs to suppress into small byte occupancy. A simple method just tries to reformat all data into binary codes. In such a way, the 90 byte data become 32 bytes for transmission. This process loses transformation time but reduces

communication load to avoid frequency congestion and data conflict to result in better transceiver efficiency. After the format change, the communication efficiency in quite improved.

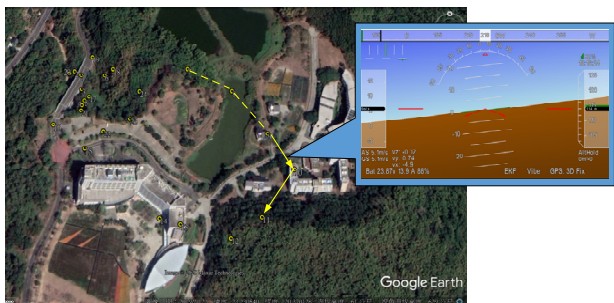

**Figure 18.** UTM surveillance data with six degrees of freedom (6DoF).

*4.7. Controller to Pilot Communication (CPC)*

A controller to pilot communication (CPC) mechanism is established following the similar concept of ATM. Mobile communication and Zello on cell phones are adopted in redundancy. Mobile communication is most widely covered in Taiwan. With the same availability, Zello can also be used [16]. In the UTM system infrastructure, a host Zello is setup on a PC as a main server for the UTM controller. The demands of CPC for the UTM Controller are multiple such as: (1) DAA avoidance, (2) Warning of no-fly zone (NFZ) or geofensing violation, (3) Re-routing clearance, (4) Lost signal.

Mobile communication used for CPC is a P2P communication; while Zello will be controller to broadcast and give the commands. Zello may wait for pilots to reply by group. Communication protocols and phraseology need to be established for all pilots by separate groups before flight. Zello is starting from small group by pilots to UTM controller. In flight operation, the controller will pull the particular pilot into the broadcasting group. After these particular pilots have landed their UAVs, their Zello groups will be dismissed to make communication clear. Since Zello occupies little frequency bandwidth, it is estimated with capability of a few hundred pilots in manipulation.

The Zello communication has a significant delay of about 2~5 s; while mobile communication may also experience 0.5~1 s. Both delays can be overcome by larger tolerance if DAA is performed. In DAA, the traffic advisory (TA) and resolution advisory (RA) are set 48 and 25 s, correspondingly. The time tolerance is enough for UTM controller to communicate to pilot to command an avoidance reaction.

During the preliminary flight tests, Zello showed a very good performance to communicate with four flight groups at different locations.

## 5. Conclusions

This paper introduces the design concept of the hierarchical UTM in Taiwan and presents trial runs of regional UTM as preliminary experiment. The introduction video is attached as Supplementary Material. The purpose of this work tries to verify the availability of ADS-B like infrastructure for UTM. In this test, APRS and LoRa are used for UAV surveillance in the Tainan Metropolitan area. Most important, this demonstrates that the Tainan RUTM is working normally for UAS surveillance. The GTS nearby tests are short flights within 8 km or less than 20 min to check with the availability of ADS-B like infrastructure. This is feasible for logistic drone delivery applications in the near future. The flight speed varies from 4 to 6 m/sec, altitude from 30 m to 90 m, AGL. The data transceiver efficiency in the nearby tests in Tables 1–5 has been analyzed by data statistics to get the binominal distribution and calculate the data variation. Five GTSs have their relative data variance.

A farther trial test was made to some remote areas in Tainan to verify and enforce the ADS-B like infrastructure in supporting UTM. Although results in Table 6 appear to be not so good, they prove that the GTS redundancy concept works well. From the analysis in Tables 1–5, the binominal distribution and data variance match with the latter test in Table 6. The preliminary test results give a direction

for further flight plans on GTS. The UTM development plan needs to work on more verification tests to establish indexes for transceiver efficiency, binominal distribution and data variance for GTSs and OBUs.

This paper aimed to focus on the feasibility analysis of UTM system performance using ADS-B like communication for UAVs. The LoRa OBU and APRS OBU are designed and fabricated. The tests verify the successful operation after GTS deployment using ADS-B Like infrastructure concept for UAV surveillance in Tainan City. This is the operation of Tainan RUTM. Referring to Scardina [6] and Orrell [8], the proposed ADS-B like infrastructure using LoRa and APRS can be feasible to invest and deploy in certain areas like Taiwan. ADS-B like systems in taking the place of ADS-B will be more adaptable and valuable for UTM.

This paper brings in the issue of surveillance technology to be the most important part to open UAS in regular commercial services. Although in the UTM concept the very low level (VLL) below 400 feet is less than the U-Space concept of 700 feet, the proposed UTM may try to create a feasible corridor for parcel service flight under single flight level, instead of multiple levels [18]. The proposed UTM operation with ADS-B like OBUs may become less complicate in surveillance.

In RUTM performance, DAA is an important function to execute. The controller-to-pilot communication (CPC) must be effective to connect to the pilots when a dangerous approach is detected. Zello and mobile 4G play a reliable solution for DAA. This is also verified in the RUTM trial tests.

The newly furnished RUTM in Tainan City can demonstrate an effective UAV surveillance using ADS-B like OBU to GTS deployment. Tests show the preliminary UTM performance. In the near future more flight tests will be carried out to check the adequate setting of a GTS antenna and its environment with least interference. Both LoRa and APRS GTSs would be working on hundreds of OBU flying trials to obtain enough data for statistical analysis.

Referring to Figure 2, the next development will expand the LoRa GTS and APRS GTS to cover a wider territory in Taiwan. The development of UTM in Taiwan is moving forward to allow UAVs flying in a safe sky under complete surveillance.

**Supplementary Materials:** The following are available online at https://youtu.be/W056iZln2UY.

**Author Contributions:** For this research work, C.E.L. proposes the system concept design, organizes GTS and OBU experiment plans for verification, writes and edits this paper for Aerospace. P.-C.S. enforces statistical analysis for communication system, carries flight tests and gets distinguished result. Y.-Y.L. conducts system organization and software programming for UTM. All authors have read and agreed to the published version of the manuscript.

**Funding:** This is an academic research project supporting from the Ministry of Science and Technology under contract MOST 108-2622-E-309-001-CC1.

**Acknowledgments:** This work is supported by Ministry of Science and Technology (MOST) under contract MOST 108-2622-E-309-001-CC1 in cooperation with Chung-Hua Telecommunication.

**Conflicts of Interest:** The authors declare no conflict of interest.

## Abbreviations

| | |
|---|---|
| ADS-B | Automatic Dependent Surveillance-Broadcast |
| ADS-R | Automatic Dependent Surveillance-Re-broadcast |
| AGL | Above Ground Level |
| API | Application Program Interface |
| APRS | Automatic Packet Reporting System |
| ARINC | Aeronautical Radio, Incorporated |
| ATC | Air Traffic Control |
| ATN | Aeronautical Telecommunication Network |
| ATM | Air Traffic Management |
| BVOLS | Beyond Visual Line-of-Sight |

| CAA | Civil Aeronautical Administration, Taiwan |
| CHT | Chunghwa Telecom Co., Ltd., Taiwan |
| CJCU | Chang Jung Christian University, Taiwan |
| CPC | Controller-to-Pilot Communication |
| CNS | Communication, Navigation and Surveillance |
| C2 | Command and Control |
| DAA | Detect and Avoid |
| FAA | Federal Aviation Administration |
| FOQA | Flight Operation Quality Assurance |
| GPS | Global Positioning System |
| GTS | Ground Transceiver Station |
| HAM | Amateur Radio (ham radio) |
| ICAO | International Civil Aviation Organization |
| IPP | Integrated Pilot Program |
| LoRa | Long Range Wide Area Network |
| MIS | Management Information System, CAA, Taiwan |
| MSL | Mean Sea Level |
| NASA | National Air and Space Administration |
| NAS | National Airspace System |
| NCC | National Communication Commission |
| NFZ | No-Fly Zone |
| NUTM | National Unmanned Aircraft System Traffic Management |
| OBU | On-Board Unit |
| P2P | Person to Person |
| RUTM | Regional Unmanned Aircraft System Traffic Management |
| Rx | Received Data |
| RSSI | Received Signal Strength Indicator |
| SATCOM | Satellite Communication |
| SITA | Société Internationale de Télécommunications Aéronautiques |
| TCAS | Traffic Alert and Collision Avoidance System |
| Tx | Transmitted data |
| UAS | Unmanned Aircraft System |
| UAV | Unmanned Aerial Vehicle |
| UTM | Unmanned Aircraft System Traffic Management |
| SSR | Secondary Surveillance Radar |
| sUAV | Small UAV |
| VLOS | Visual Line-of-Sight |
| 4G/LTE | 4th Generation/Long Term Evolution |

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
