# Peer review of "System Operation of Regional UTM in Taiwan"

_aerospace, doi:10.3390/aerospace7050065_

Round 1

Reviewer 1 Report

The topic and content are areas critical to the insertion of UAS into the national airspace. The description of testing that involves national, regional, and local entities is noteworthy.

Acronyms used for the first time should be identified, i.e., UAT, BTS and APRS.

Page 3, section 2 UTM System Concept begins with a required laundry list of what's required in a manuscript. Unless this is required by the publisher, this would not be a author entered information; it's a list of requirements for the paper.

Page 8, first paragraph below Figure 11, states "the red start at the bottom"; no red star is shown but a number listing the facility.

Figures, especially with the detail shown in some should be a little larger.

Paragraph 3.6, states "redundancy is threefold"; only 1 and 2 are show, there should be a 3 after the word "or".

Reviewer 2 Report

The article fits well within the scope of the journal. The work attempts to examine one of the interesting challenges currently obstructing the large scale deployment of UAVs within cities. 

The authors present some initial trial tests/experiments on a proposed UTM in Taiwan to examine and the effectiveness of using ADS-B for UAV surveillance under 400ft.

Overall the paper needs extensive editing of language and style. 

In the introduction, the description of the hierarchal UTM and the regulatory bodies can be elaborated further. Additional references to hierarchal UTM from literature could be used for eg. 

  • Barrado, C.; Boyero, M.; Brucculeri, L.; Ferrara, G.; Hately, A.; Hullah, P.; Martin-Marrero, D.; Pastor, E.; Rushton, A.P.; Volkert, A. U-Space Concept of Operations: A Key Enabler for Opening Airspace to Emerging Low-Altitude Operations. Aerospace 2020, 7, 24.
  • Samir Labib, N.; Danoy, G.; Musial, J.; Brust, M.R.; Bouvry, P. Internet of Unmanned Aerial Vehicles—A Multilayer Low-Altitude Airspace Model for Distributed UAV Traffic Management. Sensors 2019, 19, 4779.

In the second section presenting the UTM system concept, the authors should revise the text as well as define terms/abbreviations used in the figures such as FL and others. The authors should also motivate the reason behind designing a strictly centralised UTM, one question would be, how would strictly centralised UTM handle the forecasted numbers of UAV operations that greatly exceed those of manned civil/commercial ones.

The experimentation section needs to explain the experimental setting, methodology in more detail. The tables and results description are challenging to follow. 

Round 2

Reviewer 1 Report

I've provided an example of the Abstract with some corrections shown in red. I've also lined through unnecessary words.

The hierarchical UTM is proposed for UAS operation in Taiwan. The proposed UTM is constructed using the similar concept of ATM from transport category aviation system. Based on the airspace being divided by 400 feet of altitude, the RUTM (Regional UTM) is managed by the local government and the NUTM (National UTM) by the CAA. Under construction of UTM system infrastructure, this trial test tries to examines the effectiveness of UAV surveillance under 400 feet using ADS-B Like on-board units (OBU). The ground transceiver station (GTS) is designed with the adoptable systems. In these implementation tests, five LoRa Gateways and one APRS I-Gates are deployed to cover the Tainan Metropolitan area. The data rates are tuned in different systems from 8~12 seconds to prevent from data conflict or stream congestion. The signal coverage, time delay, data distribution, and data variance in communication are recorded and analyzed for RUTM operation. Data streaming and Internet manipulation are verified with cloud system stability and availability. Simple operational procedures are defined with priority for Detect and Avoid (DAA) for UAVs. Mobile communication and Zello broadcasts are introduced and applied to establish controller-to-pilot communication (CPC) for DAA. The UAV flight tests are generally beyond visual line-of-sight (BVLOS) near suburban areas with flight distances to 8 km. On the GTS deployment, six test locations  examine  communication coverage and effectiveness using ADS-B Like units. In system verification, the proposed ADS-B Like unitworks well in the UTM infrastructure. The system feasibility is proven with support of receiving data analysis and transceiver efficiency. The trial test supports RUTM in Taiwan for UAV operations.

There are still numerous errors where singular words should be plural, adjectives need to be added before a number of the nouns shown in the document.

I can see that the authors have done considerable updates and corrections and it is a much better product. Also, in my experience the acronyms/abbreviations are listed in the beginning of a document, just after the table of contents.

Author Response

Thanks for the valuable comments.

We made the corrections on the abstract.

However, we would like to send to professional editing service to improve this paper, if we can have allowable time.

Actually we tried our best to fulfill the basic requirement and paper quality.

Thank you.

Reviewer 2 Report

I would like to thank the authors for their elaborate responses and the revised version of the manuscript.

All comments have been properly and adequately addressed and significant changes have been done accordingly.

Author Response

Thank you for the comments.

We tried our best to fulfill the basic requirement and quality of this work.

We made some revisions according to Reviewer's comments.